

# Monensin may inhibit melanoma by regulating the selection between differentiation and stemness of melanoma stem cells

Haoran Xin[1,2], Jie Li[2], Hao Zhang[3], Yuhong Li[4], Shuo Zeng[5], Zhi Wang[4], Zhihui Zhang[1] and Fang Deng[2]

[1] Department of Cardiology, Southwest Hospital, The Third Military Medical University (Army Medical University), Chongqing, China
[2] Department of Pathologic Physiology, College of High Altitude Military Medicine, The Third Military Medical University (Army Medical University), Chongqing, China
[3] People's Liberation Army of China -32137, Zhangjiakou, Hebei Province, China
[4] Department of Cell Biology, The Third Military Medical University (Army Medical University), Chongqing, China
[5] Career Education Center, The Third Military Medical University (Army Medical University), Chongqing, China

Corresponding authors
Zhihui Zhang, xyzpj@126.com
Fang Deng, celldf@tmmu.edu.com, celldf@126.com

## ABSTRACT

Melanoma is the most lethal cutaneous malignancy that threatens human lives. Poor sensitivity to chemotherapy drugs and the high rate of resistance are the bottlenecks of melanoma treatment. Thus, new chemotherapy drugs are needed. Drug repurposing is a safe, economical and timesaving way to explore new chemotherapy for diseases. Here, we investigated the possibility of repurposing the antibiotic monensin as an anti-melanoma agent. Using three human melanoma cells and two nomal human cell lines as cell models, we found that monensin is obviously toxic to human melanoma cells while safe to nomal human cells. It effectively inhibited cell proliferation and viability, while promoted apoptosis and differentiation of human melanoma cells *in vitro*. By establishment of an animal model of transplanted human melanoma in nude mice, we demonstrated that monensin suppressed the growth of xenografts *in vivo*. At the same time, we found that melanogenesis increased and the ability of sphere and cloning forming of melanoma decreased under the treatment of monensin. Further detection about differentiation and pluripotent regulations were executed. Our results suggest that monensin is a potent inhibitor of melanoma, and its anti-tumor mechanism may be through promoting the final differentiation of melanoma stem cells and inhibiting their stemness maintenance.

## INTRODUCTION

Melanoma is a highly malignant tumor, with mortality as high as 80% (*Cummins et al., 2006*). As the early symptoms of melanoma are not obvious, most patients have been diagnosed in the middle and late stages. Melanoma is not sensitive to radiotherapy, and the treatment is mainly dependent on chemotherapy. However, poor sensitivity

to chemotherapy drugs and easy resistance are the bottleneck of melanoma treatment. The front-line clinical anticancer agents used for melanoma are mainly in five categories, including alkylating agents, anti-CTLA4 monoclonal antibodies, BRAFV600E inhibitors, C-KIT inhibitors and PD-1 inhibitors. Representative drugs include Dacarbazine, Ipilimumab, Vemurafenib, Imatinib and Nivolumab. The natural resistance rate of melanoma in order is 87.5%, 88%, 70% (white) ∼85% (yellow), 98.8% (white) ∼89.2% (yellow), and 74% (*Guo et al., 2012*; *Wu et al., 2014*). Therefore, there is an urgent need to develop effective new drugs. However, An invention of a new drug usually faces a long research period, large risk of fail and biosafty problems.

In recent years, drug reposition has attracted increasing attention. For their short development period, high biosafety, low cost and known side effects. Drug reposition has become a new hot spot in the field of cancer treatment. Monensin is secreted by the bacteria *Streptomyces cinnamonensis* (*Pressman, 1968*), and it is used to kill coccidia parasites and improve the feed conversion rate of ruminant animals. It has been reported that monensin shows a good therapeutic effect in a variety of tumors, including ovarian cancer, colon cancer, myeloma and lymphoma (*Deng et al., 2015*; *Park et al., 2003a*; *Park et al., 2003b*; *Park et al., 2002*). However, it remains unclear whether monensin has anticancer effects on human melanoma cells.

To explore the possibility of anti-melanoma effect of monensin, *in vitro*, we examined the effects of monensin on proliferation, and apoptosis of several human melanoma cell lines. *In vivo*, the effects of monensin on tumor growth was investigated by xenograft animal model. We found that monensin efficiently suppressed human melanoma at a biosafe dose, and then we discussed the foundation mechenisms of the effect.

## MATERIALS & METHODS

### Cell culture and drug

Human melanoma cells A375, Mel-624, Mel-888, Human embryonic kidney cells HEK-293 and Human bladder epithelium immortalized cells SV-HUC-1 were all purchased from the American Type Culture Collection (ATCC, Manassas, VA, USA). The cells were cultured in complete DMEM containing 10% fetal bovine serum (FBS, Invitrogen, Carlsbad, CA, USA), 100 units of penicillin and 100 µg of streptomycin, at 37 °C in 5% CO2. Drug monensin sodium salt was purchased from Solarbio (Beijing, China) and dissolved in ethanol. All the procedures were in strict accordance with the Institutional Review Board of The Third Military Medical University.

### Crystal violet cell viability assay

A375, Mel-624 and Mel-888 cells were treated with varied concentrations of monensin or ethanol control, respectively. At 24 h, 48 h and 72 h after treatment, cells were washed with PBS and fixed with 0.4% paraformaldehyde at room temperature for 20 min. Then cells stained with 0.5% crystal violet (Beyotime, Shanghai, China) at room temperature for 20 min. The cells were washed with tap water and air dried before imaging. For quantification, the cells were incubated with 100% acetic acid at room temperature for 20 min with shaking. The absorbance was set at 570 nm.

## Viable cell counting assay

A375, Mel-624 and Mel-888 cells were treated with monensin at the varied concentrations or ethanol control. At 24 h, 48 h and 72 h, cells were collected by trypsin dissociation, and stained with Trypan blue (Beyotime, Shanghai, China). Unstained viable cells and total cells were counted under a bright field microscope (Nikon, Tokyo, Japan).

## Cell cycle analysis

A375, Mel-624 and Mel-888 cells were seeded in 6-well plates and treated with varied concentrations of monensin or ethanol control. At 4 h, 8 h, 12 h after treatment, cells were collected, fixed and stained with the propidium iodide (Beyotime, Shanghai, China) for 5 min. Then the cells were subjected to flow cytometry analysis using the Flow Cytometer (BD Biosciences, San Jose, CA, USA). The flow cytometry data were analyzed with the FlowJo v10.0 software.

## CCK-8 cytotoxicity assay

Cytotoxicity was assessed by using cell counting kit-8 (CCK-8; Dojindo, Tokyo, Japan). A375, Mel-624, Mel-888, HEK-293 and SV-HUC-1 cells seeded in 96-well plates were treated with varied concentrations of monensin or ethanol control for 24 h, 48 h and 72 h. 10 µL CCK-8 reagent was added to each well, followed by an incubation at 37 °C for 60 min and reading at 450 nm using the microplate reader (Bio-RAD, Hercules, CA, USA).

## Hoechst 33258 staining

A375, Mel-624 and Mel-888 cells were seed in a 6-well plate with cover glasses respectively and treated with varied concentrations of monensin or ethanol control. At 8 h, 12 h and 16 h after treatment, cells were fixed and stained with Hoechst Staining Kit (Beyotime, Shanghai, China). Apoptotic cells were examined under a fluorescence microscope. The average number of apoptotic cells was calculated in at least ten random fields at $200\times$ magnification for each assay.

## Annexin V-FITC flow cytometry assay

A375 , Mel-624 and Mel-888 cells were seeded in 6-well plates respectively and treated with varied concentrations of monensin or ethanol control. At 4 h, 8 h and 12 h after treatment, cells were dissociated with trypsinization, washed with PBS, and resuspended in Annexin V Binding Buffer at a density of $10^6$ cells/ml. Then the cells were stained with Annexin V-FITC (BD Pharmingen, San Jose, CA, USA) for half an hour, followed by counterstaining with propidium iodide for 15 min at room temperature. After wash, the cells were subjected to flow cytometry analysis using the BD FACSCalibur-HTS. Data were analyzed by using the FlowJo v10.0 software. Each assay was done in triplicate.

## Xenograft of human melanoma cells

The use and care of animals were approved by the Laboratory Animal Welfare and Ethics Committee Of the Third Military Medical University (Approval Number SYXK (Chongqing) 20170002). A375 stably labeled with firefly luciferase (A375-FLuc) was constructed with piggyBac system (*Chen et al., 2015*; *Wang et al., 2014*; *Wen et al., 2014*). A375-Luc cells were collected and resuspended at $10^7$ cells/ml. 100 ul cells were

subcutaneously injected into the dorsal back skin of athymic nude mice (4-week-old, male, $10^6$ cells per injection, and two sites per mouse). The mice were divided into two groups ($n = 5$ per group). At three days post injection, the animals were treated with various doses of monensin (25 mg/kg or 50 mg/kg body weight) or vehicle control (ethanol) by oral administration once a day. Tumor growth was monitored by whole body bioluminescence imaging using Xenogen IVIS 200 Imaging System at days 4, 7 and 10 after injection. The mice were sacrificed at 10 days and subcutaneous tumor masses were harvested for examination.

## Sphere formation assay

A375, Mel-624 and Mel-888 cells were seeded in complete medium and placed at $10^7$ cells per 6-well Ultra Low Cluster plates (Corning) with varied concentrations of monensin or ethanol control. At 24 h, 48 h and 72 h post treatment, images were recorded at 100× magnification and the maximum diameter of cell mass were measured.

## Colony formation assay

A375, Mel-624 and Mel-888 cells were diluted in complete medium and seeded at 1000 cells per 6-well. Cells were treated with monensin at the varied concentrations and ethanol control for 24 h, then replaced the medium and continuted to culture for 6 days in DMEM supplemented with 10% FBS. After 6 days, the colonies were fixed with 0.4% paraformaldehyde for 20 min and stained with 0.5% crystal violet for 15 min. The plates were washed and visible colonies were counted and colony forming efficiency (CFE) was calculated. The colonies that were less than 2 mm in diameter or faintly stained were excluded. CFE was defined as the number of colonies divided by the number of cells seeded and expressed as percentage.

## RNA extraction and real time PCR

RNA extraction and reverse transcription were performed as previously described. Briefly, A375 cells were seeded in 6-well Ultra Low Cluster plates (Corning) with 0.4 μM monensin or ethanol control for 48 h. Total RNA was isolated using TRIzol reagents (Invitrogen). cDNA templates were generated by reverse transcription reactions with hexamer and M-MuLV reverse transcriptase (New England Biolabs, Ipswich, MA). PCR primers were designed using the Primer3 program. SYBR Green-based Real-time PCR analysis was carried out using the thermocycler Opticon II DNA Engine (Bio-Rad, CA). Relative mRNA expression was determined by normalization to the expression of a housekeeping gene, GAPDH. The Real-time PCR reactions were done in triplicate.

## Statistical analysis

Data were expressed as mean ± SD. Statistical significance of experimental results was determined by Student's $T$-test to compare the differences among two groups. For multiple group comparison, one-way ANOVA analysis of variance was performed followed by multiple comparison tests. The statistical analysis was performed using GraphPad Prism 6 (GraphPad Software, La Jolla, CA, USA). $P$ value less than 0.05 was considered as a significant difference.

## RESULTS

### Monensin is obviously toxic to human melanoma cells

To test whether monensin can decrease the livability of human melanoma, subconfluent A375, Mel-624 and Mel-888 cells were grown in increasing concentrations of monensin. Crystal violet staining results showed that cell proliferation of A375, Mel-624 and Mel-888 cells was significantly inhibited in the monensin-treated groups compared to the control group (ethanol control group), especially in A375 cells (Figs. 1A and 1B). We also conducted Trypan blue-stained after exponentially growing A375, Mel-624 and Mel-888 cells were treated with varying concentrations of monensin (0 μM to 0.4 μM). The number of viable cells decreased significantly when the concentration of monensin was increased in the three cell lines at all examined time points, especially at 72 h (Figs. 1B–1E). We also performed cell cycle analysis by using flow cytometry of monensin-treated A375, Mel-624 and Mel-888 cells. The number of cells arrested in G1 phase was significantly increased in monensin-treated cells, whereas the number of cells in S/G2/M phase was significantly decreased in monensin-treated melanoma cells, compared to the controls ($P$ value of A375 = 0.002, $P$ value of Mel-624 = 0.008, $P$ value of Mel-888 = 0.0002) (Figs. 1F and 1I). These results suggest that monensin inhibits melanoma cell proliferation, and the inhibition effect was dose-dependent.

### Monensin is non-cytotoxic to normal control human cells at the same dose

We have already know that 0 μM to 0.4 μM monensin was cytotxic to melanoma cells, if we want to use this dose to treat melanoma, side effect should be concerned. So we tested whether monensin is cytotoxic to normal control human cells at the same doses. Subconfluent human embryonic kidney cells HEK-293 and human bladder epithelium immortalized cells SV-HUC-1 were grown in increasing concentrations of monensin (0 μM to 0.4 μM). Microscopy images of HEK-293 cells and SV-HUC-1 cells showed no significant cell number decrease or morphology changes in the monensin-treated group compared to the control groups (Fig. 2B). CCK-8 cytotoxicity assay also showed that monensin was non-cytotoxic to HEK-293 cells and SV-HUC-1 cells at the indicated concentrations (IC50 cannot be obtained, Fig. 2C). However, statistical analysis of CCK-8 cytotoxicity assay of three melanoma cells showed that monensin inhibited cell activity and was significantly cytotoxic to A375 (IC50 = 0.16 μM), Mel-624 (IC50 = 0.71 μM) and Mel-888 (IC50 = 0.12 μM) (Fig. 2A). Taken together, our results demonstrate that monensin is significantly cytotoxic to melanoma cells but non-cytotoxic to normal control human cells at the same dose.

### Monensin induces apoptosis of human melanoma cells

Apoptosis may be closely related to the cytotoxic effect of monensin, therefore, we examined cell apoptosis after A375, Mel-624 and Mel-888 cells were treated with 0.4 μM monensin. Hoechst 33258 staining results revealed that the percentage of apoptotic cells was significantly increased in monensin-treated A375, Mel-624 and Mel-888 cells (Figs. 3A–3D) at 16 h after drug treatment, compared to the control groups (green arrows,

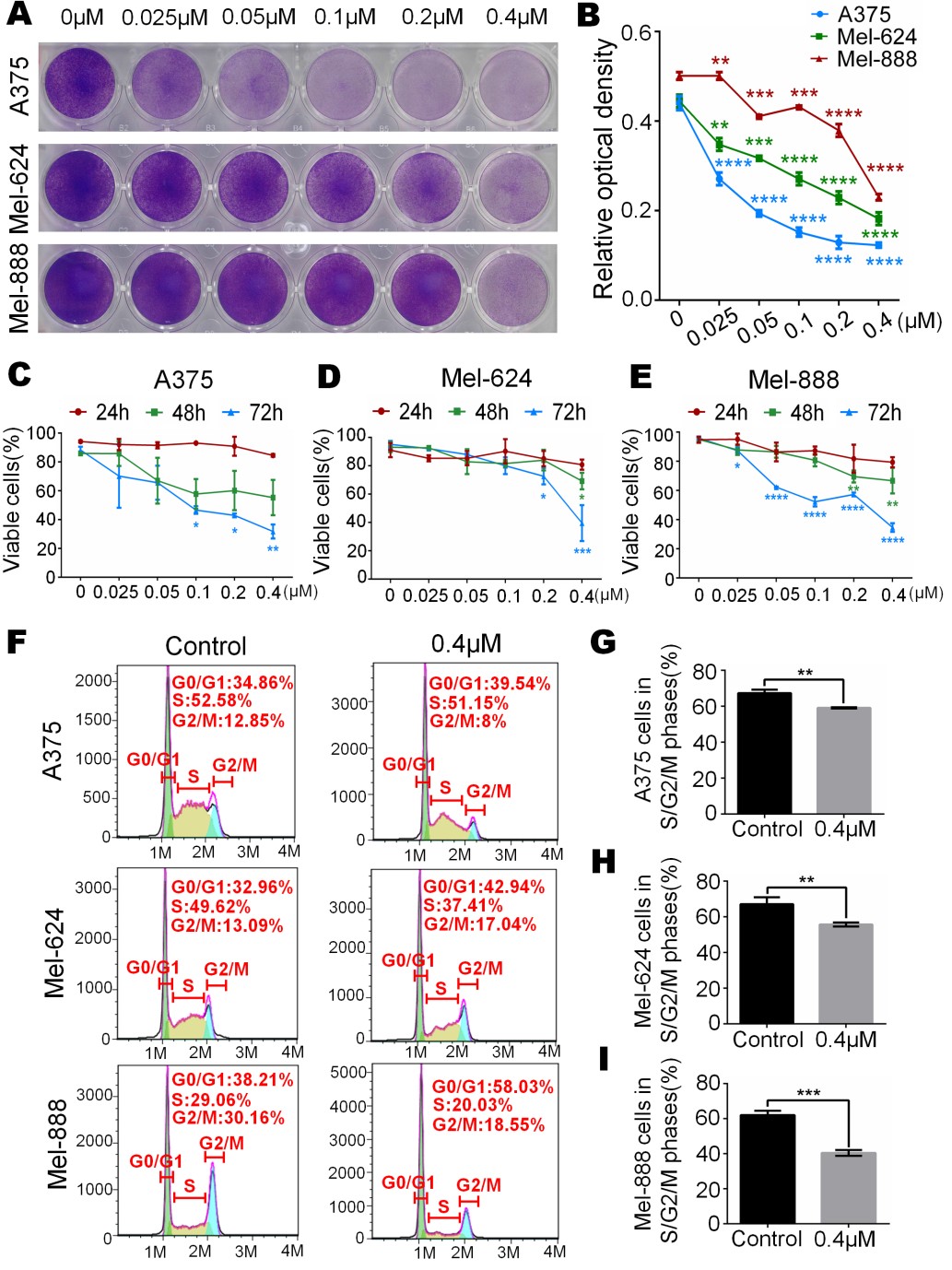

**Figure 1** **Monensin is obviously toxic to human melanoma cells.** (A) Crystal violet staining revealed that there were fewer live cells in melanoma cells A375, Mel-624 and Mel-888 treated with monensin at the indicated concentrations for 72 h, compared to the control groups. (B) Quantitative analysis of the Crystal violet-stained cells revealed that there were significantly fewer live cells in melanoma cells treated with monensin at the indicated concentrations for 72 h, compared to the control groups. (C) Quantitative analysis of Trypan blue-stained cells showed that there were fewer viable cells in melanoma cells A375 treated with monensin at the indicated concentrations for 24 h, 48 h and 72 h, compared to the control groups. 

**Figure 1 (…continued)**
(D) Quantitative analysis of Trypan blue-stained cells showed that there were fewer viable cells in melanoma cells Mel-624 treated with monensin at the indicated concentrations for 24 h, 48 h and 72 h, compared to the control groups. (E) Quantitative analysis of Trypan blue-stained cells showed that there were fewer viable cells in melanoma cells Mel-888 treated with monensin at the indicated concentrations for 24 h, 48 h and 72 h, compared to the control groups. (F) Cell cycle analysis showed that there were fewer cells in S/G2/M phase in monensin-treated groups, compared to the control groups. (G) Statistical analysis of cell cycle study showed that there were significantly fewer cells in S/G2/M phase in monensin-treated A375 cells at 12 h after treatment, compared to the control groups. (H) Statistical analysis of cell cycle study showed that there were significantly fewer cells in S/G2/M phase in monensin-treated Mel-624 cells at 12 h after treatment, compared to the control groups. (I) Statistical analysis of cell cycle study showed that there were significantly fewer cells in S/G2/M phase in monensin-treated Mel-888 cells at 12 h after treatment, compared to the control groups. $*p < 0.05$; $**p < 0.01$; $***p < 0.001$; $****p < 0.0001$.

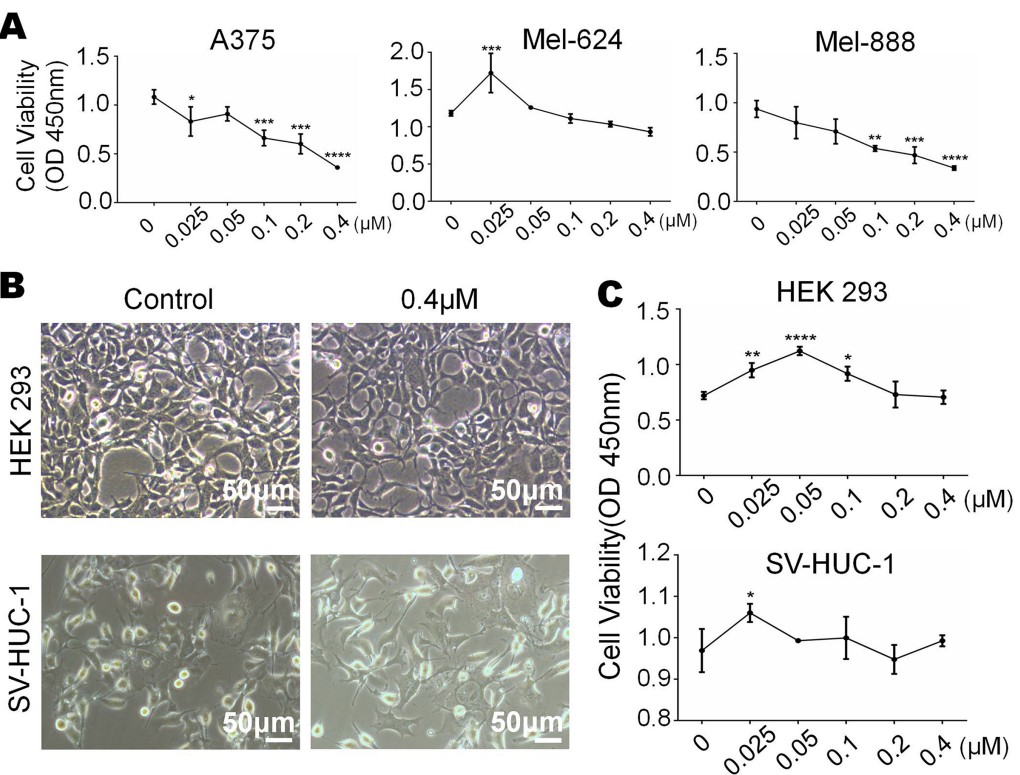

**Figure 2** **Monensin is non-cytotoxic to normal control human cells at the same dose.** (A) CCK-8 cytotoxicity assay revealed fewer live cells in melanoma cells A375, Mel-624 and Mel-888 treated with monensin at the indicated concentrations for 72 h, compared to the control groups. Absorbance, 450 nm. Each assay was done in triplicate. (B) Microscopy images revealed no decrease of HEK-293 and SV-HUC-1 treated with 0.4 μM monensin for 72 h, compared to the control groups. (C) CCK-8 cytotoxicity assay revealed no significant decrease of HEK-293 and SV-HUC-1 treated with monensin at the indicated concentrations for 72 h, compared to the control groups. Absorbance, 450 nm. $*p < 0.05$; $**p < 0.01$; $***p < 0.001$; $****p < 0.0001$.

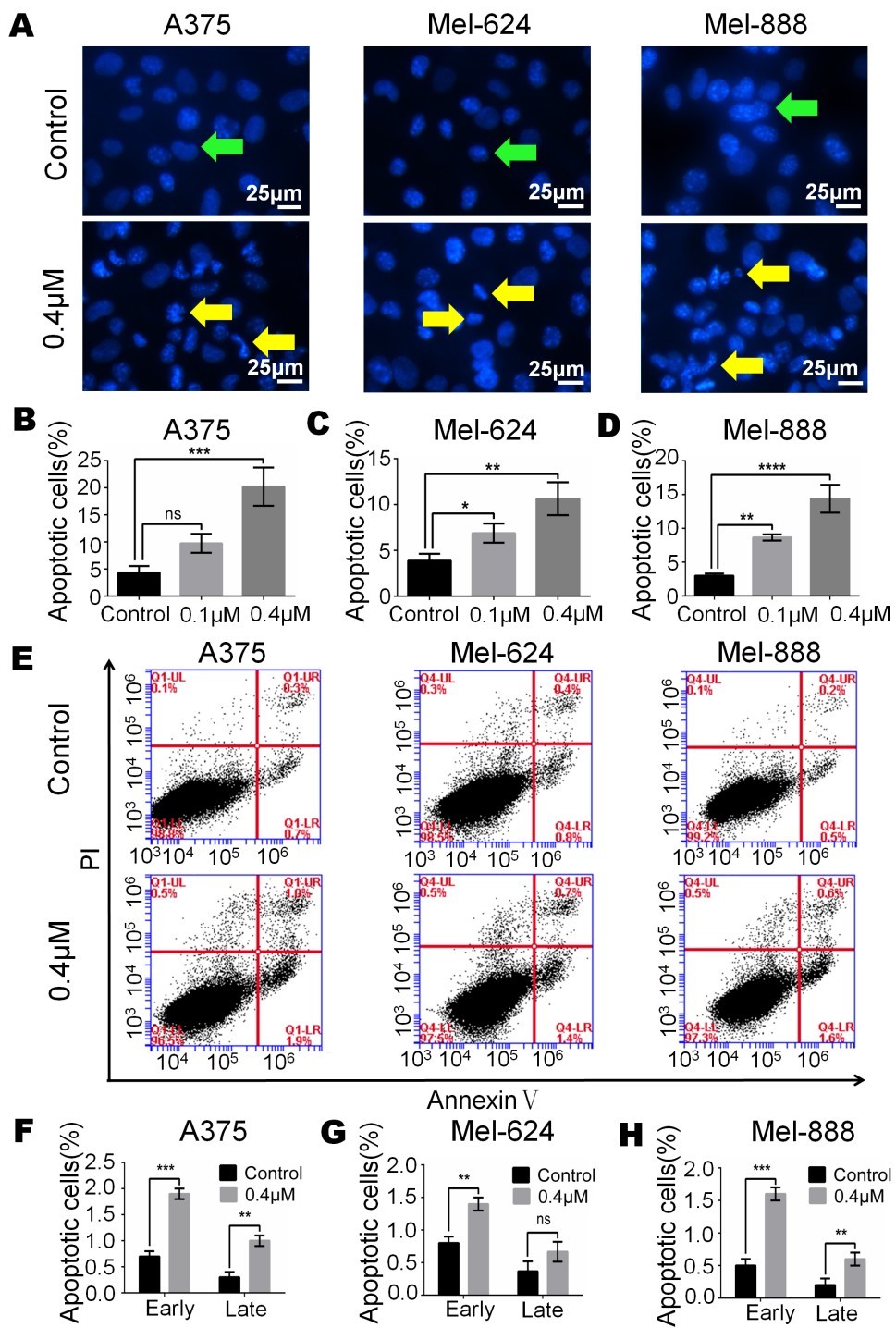

**Figure 3** **Monensin induces apoptosis of human melanoma cells.** (A) Hoechst 33258 staining revealed that there were significant more apoptotic cells in monensin-treated groups with 0.4 μM at 16 h post treatment, compared to the control groups. Green arrows, live cells; yellow arrows, apoptotic cells. (B) Statistical analysis of A375 cells revealed that there were significant more late apoptotic cells in monensin-treated group at 16 h, compared to the control groups. (C) Statistical analysis of Mel-624 cells revealed that there were significant more late apoptotic cells in monensin-treated group at 16 h, compared to the control groups. 

**Figure 3 (…continued)**
(D) Statistical analysis of Mel-888 cells revealed that there were significant more late apoptotic cells in monensin-treated group at 16 h, compared to the control groups. (E) Annexin-V apoptosis assay. A375, Mel-624 and Mel-888 cells were treated with 0.4 μM monensin, respectively. At 12 h post treatment, cells were collected and stained with Annexin V-FITC and propidium iodide, and were subjected to flow cytometry. Average percentages of apoptotic cells were calculated. (F) Statistical analysis revealed that there were more early and late apoptotic cells in monensin-treated A375 cells, compared to the control groups. (G) Statistical analysis revealed that there were more early and late apoptotic cells in monensin-treated Mel-624 cells, compared to the control groups. (H) Statistical analysis revealed that there were more early and late apoptotic cells in monensin-treated Mel-888 cells, compared to the control groups. Each assay was done in triplicate. $^*p < 0.05$; $^{**}p < 0.01$; $^{***}p < 0.001$; $^{****}p < 0.0001$; ns, no significant difference.

live cells; yellow arrows, apoptotic cells). We also checked cell apoptosis by flow cytometry (Fig. 3E). Statistical analysis revealed the proportion of Annexin V+/PI- early apoptotic cells and Annexin V+/PI+ late apoptotic cells were both increased in the monensin-treated A375 (early 1.9%, late 1%), Mel-624 (early 1.4%, late 0.6%) and Mel-888 (early 1.6%, late 0.6%) cells compared to the control group A375 (early 0.7%, late 0.3%), Mel-624 (early 0.7%, late 0.4%) and Mel-888 (early 0.5%, late 0.2%) cells (Figs. 3F–3H). Together, these results suggest that monensin can induce apoptosis in the human melanoma cell lines A375, Mel-624 and Mel-888.

## Monsensin effectively inhibits tumor growth in a xenograft model of human melanoma cells

In view of the obvious inhibitory effect of monensin on melanoma observed at the cell culture level, we are full of expectations about whether it can inhibit the growth of melanoma *in vivo*. Firefly luciferase-tagged A375 cells were subcutaneously injected into the dorsal back skin of athymic nude mice. At three days post-injection, the animals were treated with various doses of monensin (25 mg/kg or 50 mg/kg body weight) or vehicle control (ethanol) by oral administration once a day. Ten days after injection, the xenografts were collected. Tumor growth was examined using xenogen bioluminescence imaging 4, 7 and 10 days after cell injection (Fig. 4A). Quantitative analysis of the xenogen imaging data revealed that the xenografts that were treated with monensin showed significantly lower luciferase activity compared to that of the control groups (Fig. 4B). The xenografts that were treated with monensin formed significantly smaller tumors in weight compared to the control groups (Fig. 4C). These studies confirmed that monensin can effectively suppress melanoma growth.

## Monensin caninduce terminal differentiation and inhibit pluripotency of melanoma stem cells

In our study, we observed that melanin granules increased in monensin-treated groups under a phase-contrast microscope (Fig. 5A). Further, we tested the expression of tyrosinase, and the results revealed that the tyrosinase expression of A375, Mel-624 and Mel-888 cells was significantly increased in monensin-treated groups compared to the control groups (Fig. 5B). The increase in melanin granules is a manifestation of terminal differentiation in melanoma. These results inspired us the anticancer ability may be correlated with the cell fate choice between differentiation and pluripotency. So we tested cell pluripotency

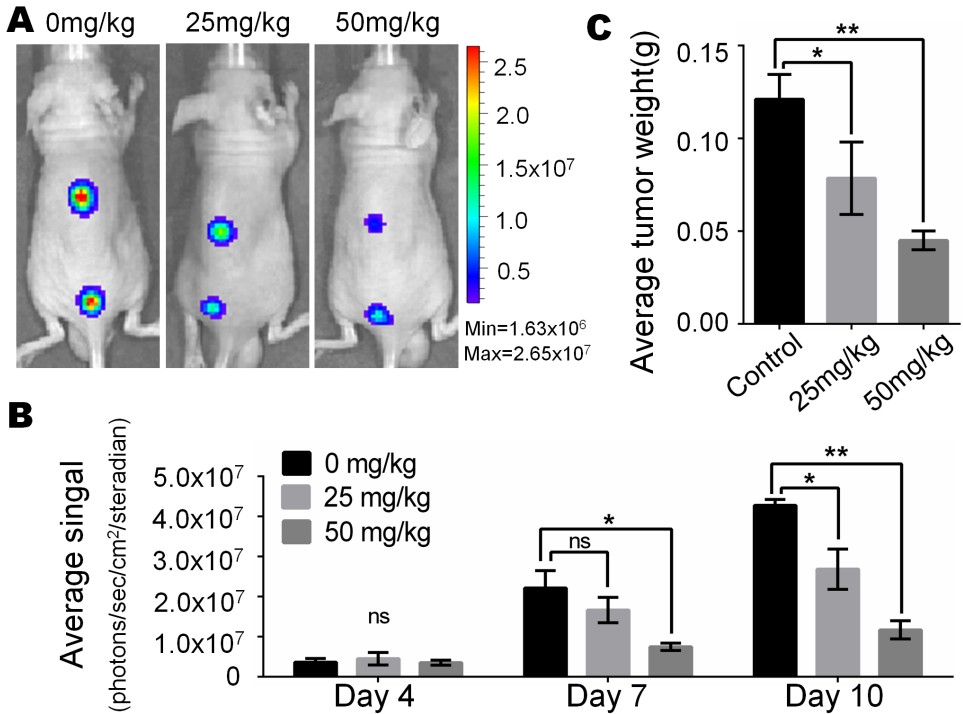

**Figure 4** **Monsensin effectively inhibits tumor growth in a xenograft model of human melanoma cells.**
(A) Xenogen bioluminescence imaging of xenograft tumor growth. Firefly luciferase-labeled A375 cells
were injected into athymic nude mice subcutaneously. The animals were treated with 0 mg/kg, 25 mg/kg
and 50 mg/kg monensin by oral administration once a day. The mice were imaged at 4, 7 and 10 days af-
ter cell injection. Representative images at day 7 are shown. (B) The average signal for each group at differ-
ent time points were calculated using the Xenogen Living Image analysis software. (C) The average tumor
weight for each group. $^*p < 0.05$; $^{**}p < 0.01$; ns, no significant difference.

maintenance after monensin treatment. The microscope images and statistical analysis
of sphere formation assay revealed that the sphere formation ability of A375, Mel-624
and Mel-888 cells was significantly decreased in monensin-treated group, compared to the
control groups (*P* value of A375 = 0.0146, *P* value of Mel-624 = 0.0004, *P* value of Mel-888
= 0.0178) (Figs. 5C and 5D). Then, we also tested the pluripotency of the three melanoma
cell lines by colony formation assay. The results revealed that monensin can significantly
inhibit the proliferation of melanoma, and this effect was shown at a low concentration of
just 0.025 μM (Figs. 5E and 5F). Sphere formation assays and colony formation assays both
suggest that monensin can inhibit the pluripotency of melanoma. Furthermore, markers
of terminal differentiation and pluripotency maintenance were detected by real time PCR.
The results showed TRP2 (Tyrosinase-related protein 2) and Sox10 (SRY-box 10 protein)
were decreased, while TRP1(Tyrosinase-related protein 1) was up-regulated (Figs. 5G–5I).
Taken together, these results strongly suggested that monensin may accelerate terminal
differentiation of melanoma stem cells and inhibit their pluripotency.

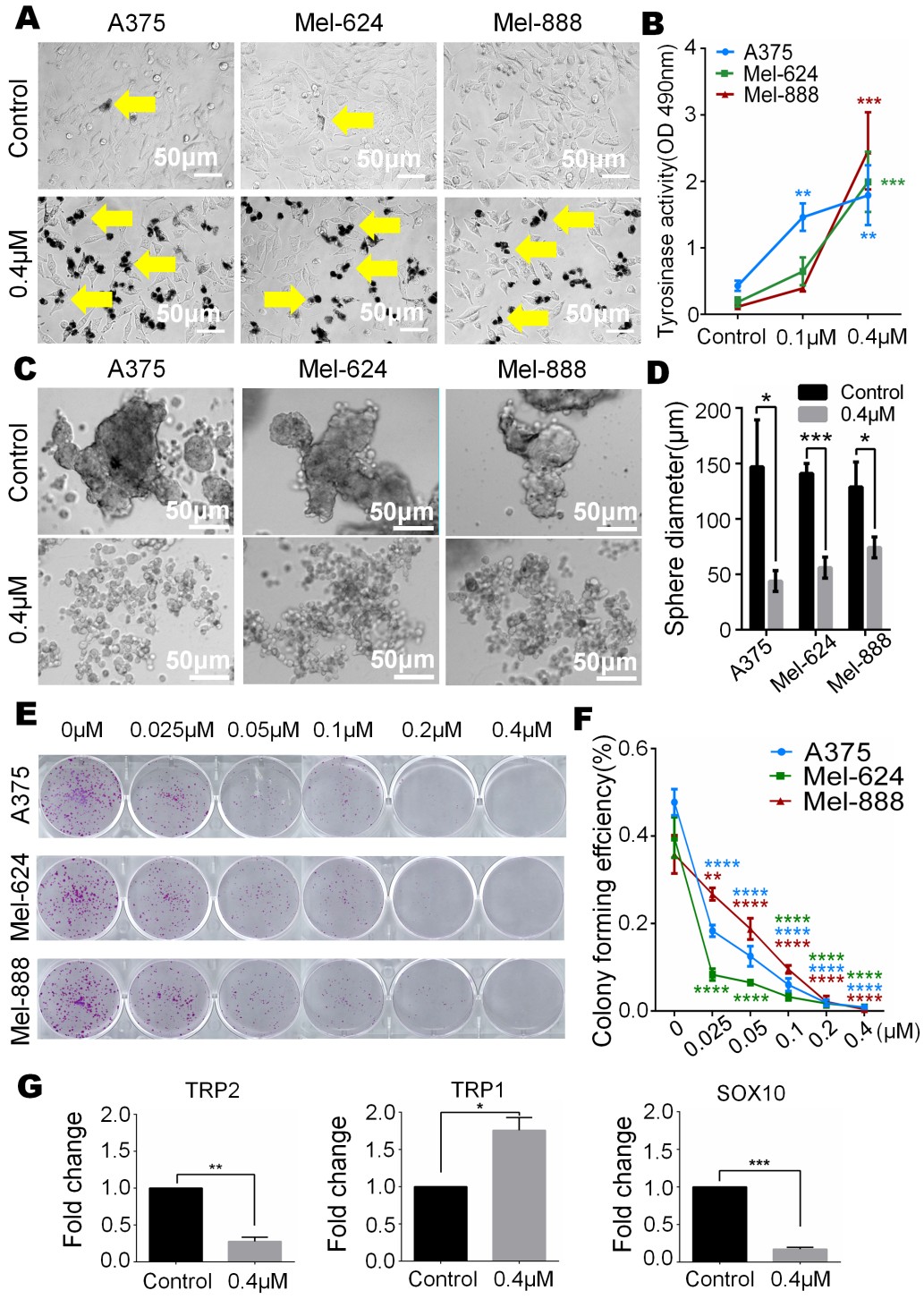

**Figure 5** **Monensin caninduce terminal differentiation and inhibit pluripotency of melanoma stem cells.** (A) Phase-contrast microscopy images of melanoma cells A375, Mel-624 and Mel-888 revealed melaningranules were increased in monensin-treated groups, compared to the control groups. Yellow arrows, melaningranules. (B) Quantitative analysis of GLuc reporters of tyrosinase showed tyrosinase expression of A375, Mel-624 and Mel-888 cells was increased with the increasing concentrations of monensin after treatment for 72 h . Absorbance, 490 nm. (continued on next page...)

**Figure 5 (…continued)**
(C) Sphere formation assay reveal ed sphere formation ability of A375, Mel-624 and Mel-888 cells was decreased in monensin-treated groups, compared to the control groups. Representative images at 48 h are shown. (D) Statistical analysis of sphere formation assay at 48 h showed that the sphere diameters are significantly decreased in monensin-treated groups, compared to the control groups. (E) Colony formation assay revealed the total number of colonies was fewer with the increasing concentrations of monensin after treatment for 6 days. Representative images are shown. (F) Quantitative analysis of colony formation assay showed colony forming effciency was significantly decreased in monensin-treated groups, compared to the control groups. (G) Real time PCR analysis for TRP2. The mRNA of TRP2 was significantly decreased in monensin-treated groups, compared to the control groups. (H) Real time PCR analysis for TRP1. The mRNA of TRP1 was significantly increased in monensin-treated groups, compared to the control groups. (I) Real time PCR analysis for SOX10. The mRNA of SOX10 was significantly decreased in monensin-treated groups, compared to the control groups.*$p < 0.05$; **$p < 0.01$; ***$p < 0.001$; ****$p < 0.0001$; ns, no significant difference.

## DISCUSSION

### Monensin may be repurposed as an effective anticancer agent for human melanoma

At an early stage, melanoma can be cured by surgery. The chemotherapy is one of late-stage treatment, however, the effect of chemotherapy is not satisfied. The chemotherapy of melanoma depends on four genetic types: including mutant BRAF, mutant RAS (N/H/K), mutant NF1, and Triple wild-type (*Amann et al., 2017*). Although BRAFV600E inhibitor was the most widely used drugs for melanoma treatment, easy resistance still limits the clinical effect seriously. The most common cause of drug resistance is MAPK/ERK pathway reactivation (*Griffin et al., 2017*). But the therapeutic effect of MEK inhibitor which can inhibit MAPK/ERK pathway is not as effective as expected (*Gupta et al., 2014*). And the targeted drugs for the other three genetic types are still in the research stage. Thus, there is a critical need to develop more effective and novel therapies to treat melanoma. Our results have demonstrated that monensin has efficient antitumor activity and effectively inhibits cell proliferation, cell viability and pluripotency, and it promotes apoptosis and differentiation of human melanoma cells.

Monensin is FDA-approved for veterinary use (beef cattle, dairy cattle, and chickens), and it is used to kill coccidia parasites and improves the feed conversion rate of ruminant animals. The *in vivo* dose of monensin we used in this study for its anticancer activity was much less than the maximum dose (200 mg/herd/day) for the prevention and control of coccidiosis (*Deng et al., 2015*). Our results have demonstrated that monensin is non-cytotoxic to normal control human cells HEK-293 and SV-HUC-1 at the same dose we used to treat melanoma cells (0 μM to 0.4 μM). These results reveal that monensin has a favorable safety profile and acts effectively at low micromolar concentrations. Moreover, the *in vitro* dose used in this study was also much less than other tumor cells. The IC50 of monensin to melanoma cells A375, Mel-624 and Mel-888 was 0.16 μM, 0.70 μM and 0.12 μM, respectively. This is a lower dose needed to achieve its anticancer activity in other tumors (IC50 is 2.5 μM in colon cancer cells and 1 μM in myeloma cells) (*Park et al., 2003a*; *Park et al., 2003b*). This finding suggests that melanoma has a higher sensitivity to monensin.

### Monensin may exert its anticancer activity by inducing terminal differentiation and inhibiting pluripotency of melanoma stem cells

We demonstrated that monensin has an anticancer effect on human melanoma, and further investigation of the detailed mechanism is needed. In earlier studies, monensin induced apoptosis-associated changes in Bax, caspase-3, and caspase-8 (*Park et al., 2002*), elevated intracellular oxidative stress (*Ketola et al., 2010*) in several human cancer cells, or exerted effects on the intracellular trafficking and processing of endocytosis (*Nishimura et al., 2015*). In our experiments, we observed that monensin promoted the apoptosis of melanoma. We know that apoptosis often occurs after cell terminal differentiation. However, few reports suggest that monensin may target cancer cells through differentiation regulation.

Interestingly, we found that the expression of melanin granules and tyrosinase activity, two indications of terminal differentiation in melanoma stem cells, were both significantly increased. The clone and sphere formation abilities, two phenotypes of stemness maintainance of melanoma stem cells, were both significantly decreased. These results suggested the anticancer activity of monensin may be related to the shift between terminal differentiation and pluripotency of melanoma stem cells. Experiments on transcriptional and post-transcriptional levels further demonstrated our hypothesis. The down-regulation of TRP2 and Sox10, with the up-regulation of TRP1, suggested the differentiation was accelerated and the pluripotency was weakened. From the above, our study showed monensin may exert its anticancer activity by inducing terminal differentiation. These results provide a new idea that we can induce differentiation of melanoma cells to treat human melanoma, similar to what we did in acute promyelocytic leukemia (*Wang & Chen, 2008*).

## CONCLUSIONS

In this study, we investigated the potential of repurposing monensin as an anti-cancer agent for human melanoma. We found that monensin can significantly inhibit human melanoma, and the mechanism may be related to the tendency of melanoma stem cells to terminal differentiation rather than stemness maintenance. Our study provides a novel choice to the treatment of melanoma and a new clue to the molecular mechanism of tumor suppression.

## ACKNOWLEDGEMENTS

We would like to thank Professor Yun Wang and Professor Yizhan Xing (Department of Cell Biology, Third Military Medical University) for their advice on cell cycle analysis and xenograft assay.

### Funding

The reported work was supported by the National Natural Science Foundation of China (NO. 81502371 to Fang Deng and NO. 81300142 to Zhihui Zhang), and the Chongqing

Basic Science and Advanced Technology Research Project (NO. cstc2016jcyjA0333 to Fang Deng) The funders had no role in study design, data collection and analysis, decision to publish, or preparation of the manuscript.

## Grant Disclosures

The following grant information was disclosed by the authors:
National Natural Science Foundation of China: 81502371, 81300142.
Chongqing Basic Science and Advanced Technology Research Project: cstc2016jcyjA0333.

## Competing Interests

The authors declare there are no competing interests.

## Author Contributions

- Haoran Xin performed the experiments, analyzed the data, prepared figures and/or tables.
- Jie Li performed the experiments.
- Hao Zhang analyzed the data.
- Yuhong Li, Shuo Zeng and Zhi Wang contributed reagents/materials/analysis tools.
- Zhihui Zhang conceived and designed the experiments, contributed reagents/materials/analysis tools.
- Fang Deng conceived and designed the experiments, authored or reviewed drafts of the paper, approved the final draft.

## Animal Ethics

The following information was supplied relating to ethical approvals (i.e., approving body and any reference numbers):

The Laboratory Animal Welfare and Ethics Committee Of the Third Military Medical University provided full approval for this research (Approval Number SYXK (Chongqing) 20170002).

## Data Availability

Xin, Haoran; Li, Jie; Zhang, Hao; Li, Yuhong; Zeng, Shuo; Wang, Zhi; et al. (2019): Monensin may inhibit melanoma by regulating the selection between differentiation and stemness of melanoma stem cells. figshare. Dataset. https://doi.org/10.6084/m9.figshare.7665506.v1.

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
