# Peer review of "Monensin may inhibit melanoma by regulating the selection between differentiation and stemness of melanoma stem cells"

_PeerJ, doi:10.7717/peerj.7354_

## Round 0.1 · original submission · Major Revisions

Authors are advised to illustrate their findings by incorporating the molecular mechanism behind monesin in the chemotherapy of melanoma.

Reviewer 1 ·

Basic reporting

see the overall comments.

Experimental design

see the overall comments.

Validity of the findings

see the overall comments.

Additional comments

Xin H et al investigated the potential therapeutic effect of antibiotic monensin in human melanoma. The authors used three human melanoma cell lines and two normal human control lines and showed monensin was toxic to the melanoma cells but not normal control cells. Monensin was shown to effectively inhibit cell proliferation, migration, invasion and cell cycle progression, while promote apoptosis and differentiation of human melanoma cells. Using the orthotopic xenograft tumor model, the authors demonstrated that monensin effectively inhibited the growth of xenograft tumors in vivo. Mechanistically, the authors showed that monensin affected multiple cancer-related pathways, including TCF/LEF, Smad and STAT3, and induced the terminal differentiation of melanoma cells. Thus, the authors concluded that monensin may be repurposed as an anti-melanoma agent.

The reported findings are interesting and should have significant translational potential. Experimental design was straightforward and reasonably executed. Data analysis and interpretations were appropriate. Overall, the conclusion is supported by the authors’ experimental findings. However, the manuscript can be significantly improved if the following concerns are fully addressed:

1). Figure 6A revealed that monensin up-regulated several proliferation-related pathways. These results are counterintuitive considering monensin’s strong anti-proliferative activity. The authors should provide explanations of such results. Furthermore, the authors should include a constitutive reporter as a positive control to demonstrate the specificity.

2). The authors mentioned that monensin may inhibit melanoma cell growth by inducing terminal differentiation. However, the authors did not include sufficient data on this conclusion. Did the authors observe more production of melanin or upregulation of terminal differentiation markers upon monensin stimulation.

3). The title is somewhat cryptic or ambiguous, and does not reflect the overall conclusion. Thus, the title should be revised.

4). The cited references should be expanded a bit as currently available melanoma therapies should be discussed.

5). The authors should perform extensive spell checks. The manuscript would be greatly benefited from a professional language editing service. Numerous errors are presented throughout the text. For example, “celllines and two nomal human cell lines” in the Abstract. The statement like “Monensin is non-cytotoxic to common human cells at the same dose” is very confusing. For “Common human cells”, do the authors mean “normal control human cells”?

Reviewer 2 ·

Basic reporting

Language needs editing professionally. Other aspects are sufficient.

Experimental design

Solid and sufficient

Validity of the findings

Some statements need to be rephrased. See comments.

Additional comments

Xin and colleagues have investigated the potential of the antibiotic monensin as a chemotherapeutic agent for melanoma. They have shown that monensin inhibits cell proliferation, migration, invasion, induces apoptosis and markers for melanocytic differentiation, and suppresses melanoma tumor growth in mice. Using reporter assays, the authors demonstrated that monensin affects multiple cancer-related pathways, and speculated that monensin may exert its anti-melanoma effect through a differentiation like mechanism. The strength is the extensive characterization of the effect of monensin on melanoma cells in vitro in multiple readouts at the cellular levels, as well as melanoma growth in vivo. The weakness is the lack of molecular mechanism for the effect of monensin.

Specific comments:

1. Line 329-340: The statement “Monensin may target cancer cells through differentiation-related signaling pathways” is not supported by data showed. It needs to be rephrased.
2. It is unclear how the observations from the reporter assay are associated with monensin’s effect on melanoma cells. Unbiased assay such microarray or RNA-seq followed by validation may reveal the transcriptome-wide gene targets of monensin.
3. Fig. 6. Pathways identified by the candidate approach can be adaptive response, instead of a causative mechanism. This needs to be further discussed.
4. Line 317-320: Contamination from meat consumption is different from using it as a therapeutic drug, in terms of doses. Therefore, this needs to be considered in the discussion.
5. What is the rationale for the in vivo dose selection?
6. The manuscript needs to be edited for language.

---

## Round 0.2 · Minor Revisions

Authors are advised to address the remaining comments from Reviewer 2.

Reviewer 1 ·

Basic reporting

see general comments

Experimental design

see general comments.

Validity of the findings

see general comments.

Additional comments

The authors were responsive to the reviewers' comments and addressed mot if not all of the comments. The revised manuscript is significantly improved and thus recommended for consideration for acceptance.

Reviewer 2 ·

Basic reporting

The authors worked hard to address the reviewer’s comments by revising statements and adding new data. However, the revision is not satisfactory.

Specific comments:

1.Discussion-Paragraph #1, Line 259: “At an early stage, melanoma can be cured by surgery. The treatment of late-stage melanoma is dependent on chemotherapy, however, the effect of chemotherapy is not satisfied.” The statement is inaccurate. The mainstream therapy for advanced melanoma is immunotherapy. Chemotherapy is not effective. Targeted therapy such as BRAF inhibitor show resistance in majority of patience. The authors need to find a melanoma expert to edit the paper.

2. Discussion-Last paragraph: “…and inhibiting pluripotency of melanoma stem cells”. Pluripotency was not defined in the manuscript. Melanoma stem cell theory was still in controversy. The authors need to be careful using melanoma stem cells and/or pluripotency, unless data are provided to proof them. Tumorsphere assay is not sufficient.


3. Conclusion-“ We found that monensin can significantly inhibit human melanoma at a biologically safe dose”: the authors need to justify what it means a biologically safe dose.

Experimental design

OK

Validity of the findings

OK

Additional comments

See above comments.

---

## Round 0.3 · accepted · Accept

Thanks for submitting your revised manuscript. It is a pleasure to accept your manuscript in its current form for publication.

Reviewer 2 ·

Basic reporting

OK

Experimental design

OK

Validity of the findings

OK

Additional comments

The authors have addressed all my comments satisfactorily.